# Solar water splitting by photovoltaic-electrolysis with a solar-to-hydrogen efficiency over 30%

Jieyang Jia[1,*], Linsey C. Seitz[2,*], Jesse D. Benck[2,*], Yijie Huo[1], Yusi Chen[1], Jia Wei Desmond Ng[2,3], Taner Bilir[4], James S. Harris[1] & Thomas F. Jaramillo[2]

Hydrogen production via electrochemical water splitting is a promising approach for storing solar energy. For this technology to be economically competitive, it is critical to develop water splitting systems with high solar-to-hydrogen (STH) efficiencies. Here we report a photovoltaic-electrolysis system with the highest STH efficiency for any water splitting technology to date, to the best of our knowledge. Our system consists of two polymer electrolyte membrane electrolysers in series with one InGaP/GaAs/GaInNAsSb triple-junction solar cell, which produces a large-enough voltage to drive both electrolysers with no additional energy input. The solar concentration is adjusted such that the maximum power point of the photovoltaic is well matched to the operating capacity of the electrolysers to optimize the system efficiency. The system achieves a 48-h average STH efficiency of 30%. These results demonstrate the potential of photovoltaic-electrolysis systems for cost-effective solar energy storage.

[1] Department of Electrical Engineering, Stanford University, 350 Serra Mall, Stanford, California 94305, USA. [2] Department of Chemical Engineering, Stanford University, 443 Via Ortega, Shriram Center Room 305, Stanford, California 94305, USA. [3] Institute of Chemical and Engineering Sciences, Agency for Science, Technology and Research, Jurong Island 627833, Singapore. [4] Solar Junction, 401 Charcot Avenue, San Jose, California 95131, USA. * These authors contributed equally to this work. Correspondence and requests for materials should be addressed to J.S.H. (email: jharris@stanford.edu) or to T.F.J. (email: jaramillo@stanford.edu).

The sustainable nature of solar electricity along with its associated large resource potential and falling costs have motivated a rapid increase in the deployment of utility-scale solar electricity generation plants in recent years[1]. As the installed capacity of photovoltaics (PVs) continues to grow, cost-effective technologies for solar energy storage will be critical to mitigate the intermittency of the solar resource and to maintain stability of the electrical grid[2]. Hydrogen generation via solar water splitting represents a promising solution to these challenges, as $H_2$ can be stored, transported and consumed without generating harmful byproducts[3–8]. However, the cost of $H_2$ produced by electrolysis is still significantly higher than that produced by fossil fuels. The Department of Energy has calculated the $H_2$ threshold cost to be \$2.00–\$4.00 per gallon of gasoline equivalent[9], whereas the most up-to-date reported $H_2$ production cost via electrolysis is \$3.26–\$6.62 per gallon of gasoline equivalent[10]. There are several promising approaches to large-scale solar water splitting, including photochemical, photoelectrochemical (PEC) and PV-electrolysis systems[8]; none of these approaches are currently economically viable compared with today's technologies[3,6,11].

To be practical for large-scale deployment, the cost of solar $H_2$ generation must be significantly reduced. Previous studies have predicted that achieving a high solar-to-hydrogen (STH) efficiency is a significant driving force for reducing the $H_2$ generation cost[12–14]. To date, the highest efficiency demonstrated using a PEC water splitting system with at least one semiconductor–liquid junction is 12.4% (refs 15,16). Theoretical studies using a variety of assumptions have predicted that the maximum attainable efficiency using a tandem PEC water splitting device is 23–32% (refs 17–20). PV-electrolysis systems have demonstrated exceptional potential to achieve even higher STH efficiencies[8,11,21–26]. The highest STH efficiency demonstrated to date, 24.4%, was delivered by a PV-electrolysis system using GaInP/GaAs/Ge multi-junction solar cells and polymer electrolyte electrochemical cells[24]. For comparison, the best multi-junction PV created to date demonstrated a solar-to-electricity conversion efficiency of 46.0% under concentrated illumination[27]. In theory, PV-electrolysis systems could potentially achieve up to 90–95% of the PV efficiency, which could allow for PV-electrolysis efficiencies of ∼57% for a 3J cell and ∼62% for a 4J or 5J cell[28]. These values indicate that there is significant room for further improvement in the performance of PV-electrolysis system prototypes.

The discrepancy between reported STH efficiencies for PV-electrolysis devices and stand-alone solar-to-electricity PV efficiencies mainly arises from poor matching of the current–voltage (I–V) characteristics of multi-junction PVs with those of water electrolysers[8]. The maximum power-point voltage ($V_{MPP}$) of a typical commercial triple-junction solar cell is in the range of 2.0–3.5 V under 1–1,000 suns of illumination. However, the thermodynamic minimum voltage required to electrolyse water is only 1.23 V at 300 K (refs 7,29), with practical operating voltages in the range of 1.5–1.9 V (refs 7,30). Electrolysing water using a voltage in excess of the thermodynamic minimum voltage results in energy wasted as heat rather than stored in $H_2$ chemical bonds. Previously, this limitation was overcome by coupling multiple PV and/or electrolyser units in series, to optimize the match between the voltage characteristics of these device components[23,24], although the efficiencies achieved were still far from optimal.

In this work, we employ a high-efficiency triple-junction solar cell with two series-connected polymer electrolyte membrane (PEM) electrolysers to achieve very high STH efficiency. Our system produces $H_2$ with a 48 h average STH efficiency of 30%, the highest efficiency reported to date for any solar $H_2$ production system, to the best of our knowledge. This work demonstrates the potential for building extremely high-efficiency solar $H_2$ production systems using current state-of-the-art commercially available solar cells and laboratory PEM electrolysers. The device design presented herein could provide a viable route to implementing large-scale solar water splitting installations.

## Results

**PV-electrolysis system design.** A schematic of the PV-electrolysis system is shown in Fig. 1. The solar cell is a commercially available triple-junction solar cell manufactured by Solar Junction, with an active area of 0.316 cm[2]. From top to bottom, the three subcells of the PV are made of InGaP ($E_g = 1.895$ eV), GaAs ($E_g = 1.414$ eV) and GaInNAs(Sb) ($E_g = 0.965$ eV) respectively[31]. The cell was installed on a water-cooled stage, which maintained a cell temperature of ∼25 °C and was illuminated with white light from a xenon arc lamp to simulate concentrated AM 1.5D solar illumination. (See Supplementary Fig. 1 for comparative xenon arc lamp and AM 1.5D spectra.) The two PEM electrolysers consist of Nafion membranes coated with 0.5 mg cm$^{-2}$ Pt black catalyst at the cathode and 2 mg cm$^{-2}$ Ir black catalyst at the anode. The two electrolysers were connected in series with the PV cell and a potentiostat, which was used to measure the current through the circuit. Water was pumped into the anode compartment of the first electrolyser, whereas the cathode of the first electrolyser had no input flow. The water and $O_2$ effluent from the first electrolyser's anode compartment flowed into the anode compartment of the second electrolyser. Likewise, the $H_2$ from the cathode side of the first electrolyser flowed into the cathode side of the second electrolyser. The $H_2$ and $O_2$ products flowing out of the second electrolyser were collected and quantified, whereas the unreacted water was fed back into a water reservoir and recycled through the system. The temperature of the electrolysers was held at ∼80 °C, consistent with standard operating conditions for industrial water electrolysers[21,22]. The system operated continuously for 48 h without interruption. Further details of the experimental setup are included in the Methods section.

**System performance.** The I–V characteristics of the cell under both 1 sun and concentrated illumination are shown in Fig. 2. Following a standard method for characterizing concentrated PV

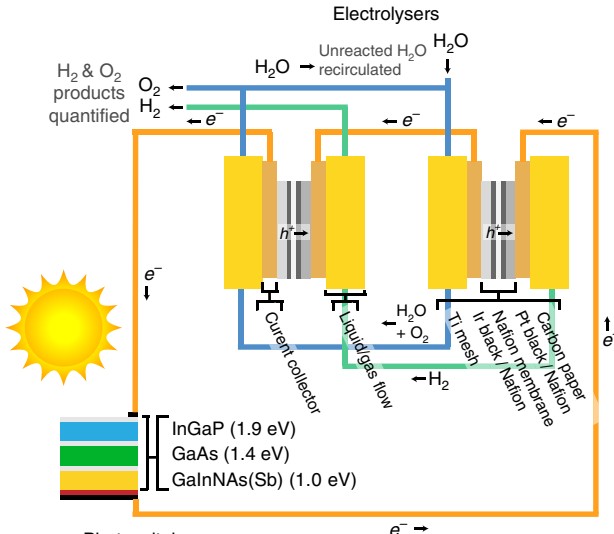

**Figure 1 | PV-electrolysis device schematic.** The PV-electrolysis system consists of a triple-junction solar cell and two PEM electrolysers connected in series.

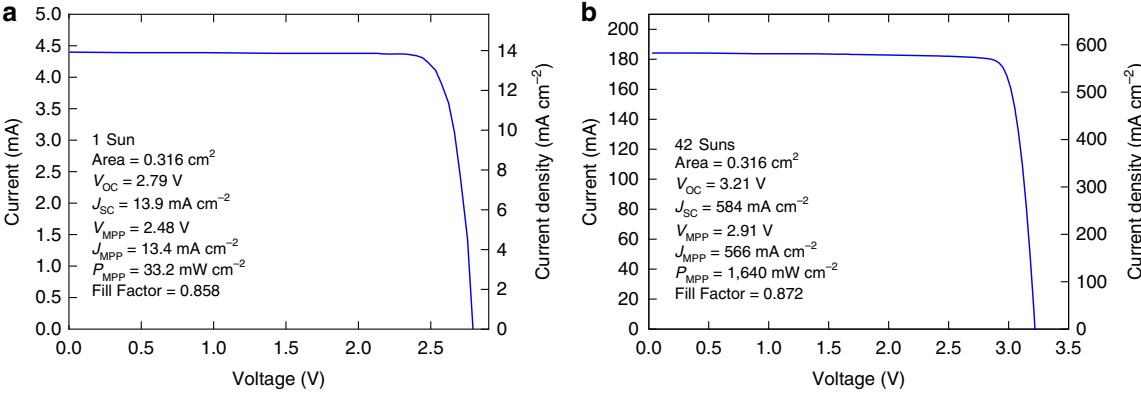

**Figure 2 | PV cell performance.** The *I–V* characteristics of the triple-junction solar cell under (**a**) 1 sun and (**b**) 42 suns, which is the illumination concentration used for the 48 h electrolysis. The key performance parameters are included in the figure. *I–V* curves were collected using both forward and backward voltage sweeps. These measurements generated identical results; thus, this figure shows only the forward sweep results.

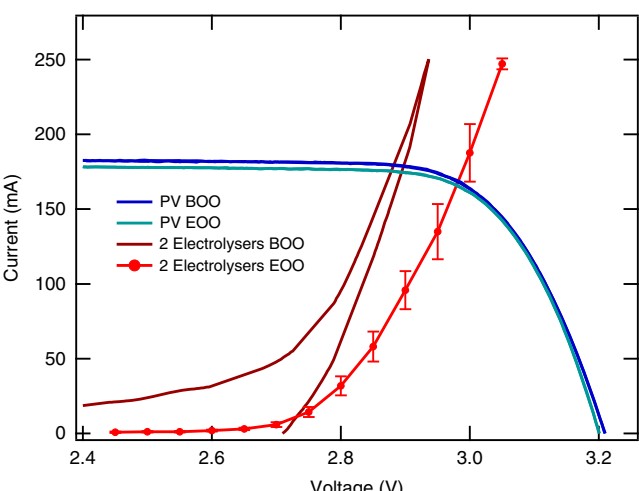

**Figure 3 | PV cell and PEM electrolyser performance at the beginning and end of operation.** The *I–V* characteristics of the triple-junction solar cell and the dual-electrolyser at both beginning-of-operation (BOO) and end-of-operation (EOO). The blue and cyan curves are the solar cell *I–V* curves under 42 suns at BOO and EOO, respectively. The dark and light red curves are the *I–V* curves of the dual-electrolysers at BOO and EOO, respectively. The BOO electrolyser *I–V* was measured as a single cyclic voltammogram to minimize catalyst degradation before system operation. The EOO electrolyser *I–V* is presented as the average current for 2 min holds at each potential, shown with error bars indicating 1 s.d. and a connecting line to guide the eye.

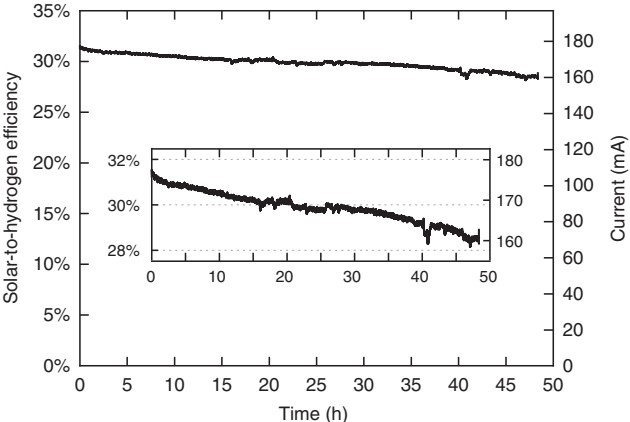

**Figure 4 | STH efficiency.** The STH efficiency of the PV-electrolysis system was measured over a 48 h continuous operation. The right vertical axis shows the current passing through the dual electrolyser and the left vertical axis shows the corresponding STH efficiency. The inset highlights a smaller *y* axis range for improved clarity.

cells, we used the ratio of the short circuit currents to calculate that the cell was illuminated with 42 suns white light under the concentration condition (calculation details provided in Supplementary Note 1)[32,33]. At the maximum power point, the cell voltage ($V_{MPP}$) output is 2.91 V and the current density ($J_{MPP}$) is 565.9 mA cm$^{-2}$. The measured solar-to-electricity efficiency is 39% under these operating conditions. The *I–V* characteristics of the dual electrolyser and the solar cell, measured before and after the 48 h operation, are shown in Fig. 3. The cross-point of a dual-electrolyser *I–V* curve and a solar cell *I–V* curve is the system-coupling point and indicates the operating voltage ($V_{OP}$) and current ($I_{OP}$) for the system. It can be seen that at the beginning of operation, $V_{OP}$ is slightly lower than $V_{MPP}$. At the end of operation, as the activity of the dual-electrolyser decreases, $V_{OP}$ is slightly higher than $V_{MPP}$. The decrease in electrolyser perfor-

mance is most likely to be due to catalyst degradation (loss of catalyst material) as well as potential membrane degradation or contaminants introduced from recirculating water for extended periods of time. In addition, small amounts of water were observed in the cathode outlet stream, suggesting water crossover and buildup on the cathode side, which could have blocked catalyst active sites and decreased hydrogen production over time. Although the increase of $V_{OP}$ over $V_{MPP}$ resulted in a slight decrease of $I_{OP}$ and the STH efficiency over the 48 h test operation, the solar cell and dual-electrolyser were well matched near the maximum power point of the solar cell, ensuring PV-electrolysis performance near the optimum.

**STH efficiency.** Figure 4 shows the electrolysis current and the corresponding STH efficiency through the 48 h experiment. The operating current decreased by only 10% over this period from an initial value of 177 mA to a final value of 160 mA. The STH efficiency was calculated by multiplying two times the thermo-dynamic potential ($V_{redox}$), the electrolysis current ($I_{WE}$) and the Faradaic efficiency for hydrogen evolution ($\eta_F$), then dividing by the input light power ($P_{in}$)[34]. The factor of two comes from the fact that there are two electrolysers connected in series; thus, the electrolysis current is driving two electrolysis reactions at the

same time.

$$\text{STH} = \frac{2 \times V_{\text{redox}} \times I_{\text{WE}} \times \eta_{\text{F}}}{P_{\text{in}}} \qquad (1)$$

In this system, the thermodynamic potential for the water-splitting reaction is 1.18 V, as the dual electrolyser was operating at 80 °C. A $P_{\text{in}}$ value of 1,328 mW was obtained by multiplying the operational concentration (42 suns) by the 1 sun illumination intensity (100 cm$^{-2}$) and the solar cell area (0.316 cm$^2$). The Faradaic efficiency of the electrolyser was measured at 20 min intervals several times throughout the 48 h experiment (photographs of the measurement apparatus are shown in Supplementary Fig. 2). The results of the Faradaic efficiency measurements and calculations indicate that the electrolyser is selective for producing H$_2$ and O$_2$ (results shown in Supplementary Fig. 3). We assume that our system has nearly 100% Faradaic efficiency for hydrogen production, consistent with other studies of PEM electrolysers[23–25]. Further details about the product quantification measurements are in Supplementary Note 2 and procedure for STH calculations are provided in Supplementary Note 3.

Based on equation (1), the system reached an average STH efficiency of over 31% in the first 20 min of operation and maintained an STH efficiency of >30% for the first 20+ h. The STH efficiency slowly decreased over 48+ h of testing and averaged ∼28% for the last 20 min of operation. The average STH efficiency achieved over the entire 48+ h of operation was 30%. The decrease in STH over 48 h of operation is mainly due to an increase of $V_{\text{OP}}$ over the maximum power point primarily caused by a decrease in performance of the electrolysers over time, an issue that is readily addressable through electrolyser development, as commercial electrolysers have demonstrated tens of thousands of hours of stable operation. Although electrolyser performance accounts for most of the decrease in PV-electrolysis efficiency over time, Fig. 3 also shows that the current of the solar cell was slightly lower at the end of operation compared with the beginning of operation, which is due to a slight decrease of the incident light intensity. This contributed approximately one percentage point to the total STH loss, as the STH calculation was performed using the initial incident light intensity; thus, the overall STH efficiencies reported are conservative values.

## Discussion

The efficiency analysis assumes that the light incident on the PV cell is the only energy input into the system. In reality, there can be additional energy inputs, including the energy required to cool the PV, heat the PEM electrolysers and pump water into the electrolysers. In a concentrated PV device illuminated by natural sunlight, there may also be optical losses in the lenses or mirrors used to concentrate the incident sunlight onto the PV cell, which could reduce the system efficiency. We have chosen to neglect these factors in this study for several reasons, among them to allow for direct comparisons with results from previous studies. We note in general that laboratory-scale systems are designed differently than commercial-scale systems and the losses are not identical. For instance, in this study, lenses or mirrors were not needed to achieve the simulated concentrated solar light; thus, there are no optical losses to measure. In addition, commercial-scale PV systems and PEM electrolysers have been designed to minimize energy losses involving cooling, heating, pumping and so on; thus, the losses in a laboratory-scale demonstration system are less relevant, in addition to being more difficult to accurately quantify. Ultimately, the methods used to supply the aforementioned additional energy inputs in a laboratory-scale system are not optimized; thus, they are not an accurate representation of what could be achieved in large-scale PV-electrolysis systems. As

a result, the STH efficiency numbers we have reported for this device represent a limiting upper bound in regard to any additional auxiliary energy inputs that would be required for any commercially scaled water-splitting system.

We note that the laboratory-scale device described in this study is constructed from components that could be scaled up to larger PV-electrolysis installations. Optimizing the PV-electrolysis system design will be crucial for enabling the deployment of economical large-scale installations. Appropriate designs will minimize the heating, cooling and pumping energy inputs. Most commercial concentrated PV modules use passive cooling methods to regulate the PV cell temperature[35] and PEM electrolysers will naturally run at temperatures well above ambient conditions, as the excess energy from overpotentials at the anode and cathode is dispersed as heat. If needed, additional heat could be supplied by the sunlight that is not converted to electricity. It might also be desirable to operate the PV at a temperature >25 °C. Although this would decrease the PV's open circuit voltage, it would increase the current output and decrease the energy input needed to cool the PV (if active cooling methods were employed). This could result in an increased STH efficiency if the PV still produced enough voltage to drive the dual-electrolyser. The PV operating temperature and other operation parameters pose tradeoffs to be optimized. Overall, the system design and operating conditions must be chosen to minimize the levelized cost of hydrogen produced over the operating lifetime of the system. Although the components used in this device are expensive, including the III–V PV fabrication and the precious metal catalysts, the very high STH efficiency achieved shows that this strategy may have the potential to produce hydrogen at low cost, in particular if lower-cost materials and fabrication methods can be developed. Technoeconomic models of large scale, centralized solar H$_2$ production facilities suggest that achieving high STH efficiency is one of the most important factors in reducing the cost of H$_2$, potentially even more important than reducing the cost of the absorber or catalyst materials[13,14]. Continued development of lower cost, higher efficiency PV cells and electrolysers, optimized PV-electrolysis system designs and technoeconomic models to predict hydrogen costs are all important subjects for continuing research.

In summary, we report a PV-electrolysis system that demonstrated an average STH efficiency of 30% over a 48 h period of continuous operation. This is the highest STH efficiency reported to date and the first solar water splitting system that demonstrates a STH efficiency reaching 30% or higher. Coupling a high-efficiency multi-junction solar cell with two electrolysers in series is an effective way to minimize the excessive voltage generated by a multi-junction solar cell, allowing for greater utilization of the high-efficiency PV for water splitting. This system also primarily uses commercially available components, suggesting that similar techniques could be implemented on a large scale. The demonstration of this high-efficiency system is an important step closer to the US Department of Energy technology and cost goals, and shows great opportunities for solar energy storage and H$_2$ production with solar water splitting.

## Methods

**Solar simulator calibration and Solar Cell characterization.** For the 1 sun measurement, we used a solar simulator (ABET Technologies, Model Sun2000) equipped with a 550 W xenon lamp as a light source. The GaInP/GaAs/GaInNAsSb multi-junction solar cell (0.316 cm$^2$ area) was manufactured by Solar Junction.

As the cells were operated under concentrated sunlight, the calibration of the solar simulator for 1 sun conditions (100 mW cm$^{-2}$) was carried out using the AM 1.5 Direct spectrum (ASTM G173). Although reference solar cells can be used to adjust a simulator for appropriate total power output, spectral control is crucial for accurate multi-junction cell measurements. For this reason, the external quantum efficiency (EQE) of each sub-junction of the multi-junction stack was measured

using a grating monochromator (Newport CS260) calibrated with silicon and germanium photodetectors (Newport 918D-UV, 918D-IR). All light sources and photodetectors were calibrated by the manufacturers before the experiment. These EQE measurements were integrated with the AM 1.5D spectrum to determine short-circuit current densities and to understand the current-limiting junction of the cells. The EQE of each sub-junction and the AM 1.5D spectrum are shown overlaid in Supplementary Fig. 1. The cells were all top-junction limited, which allowed the simulator to be tuned without luminescent coupling impacting the current[27]. The 1 sun $I$–$V$ characteristics of the cell were measured using this condition. These data are shown in Fig. 2a.

For the PV-electrolysis measurement, we used a multi-sun solar simulator (Newport, Model 66921) with a 1,000 W xenon lamp as the white light source. A water filter was applied to the light beam, to remove the excessive infrared component from the lamp spectrum and to better match the AM 1.5D spectrum. The cell package was mounted onto a water-cooling stage such that a surface temperature of 25 °C was maintained. The distance between the cell and the lamp was adjusted to achieve the desired short circuit current ($J_{SC}$). The intensity of the concentrated white light was determined from the ratio of the $J_{SC}$ under concentration to the $J_{SC}$ under 1 sun illumination, consistent with standard practices for characterizing concentrated PV cells[32,33]. No concentrator optics were placed between the simulator and the cell and therefore the possible effects of concentrator optics were not considered in efficiency calculation. As the cell was cut and installed in a standard CPV cell package, the whole cell area is active under illumination; therefore, no aperture or mask was used. Numerical modelling based on the junction ideality factor was later conducted to determine whether open-circuit voltage ($V_{OC}$) inflation due to the mismatch between the solar simulator spectrum and the AM 1.5D spectrum resulted in STH inflation[36–38]. The results of these calculations show that the lamp spectrum mismatch did not cause significant STH inflation in our experiments (details provided in Supplementary Note 4 and Supplementary Table 1).

During the duration of the experiment the solar cell performance was very stable, as expected for III–V solar cells. No hysteresis or time-transient behaviour was observed during the $I$–$V$ measurement. The $I$–$V$ characteristics results in Fig. 2 were measured with a forward voltage sweep rate of 50 mV s$^{-1}$ and a sampling period of 0.02 s. Changing the sweep rate, direction or sampling rate did not generate a noticeable difference in the results.

**Electrolyser fabrication and characterization.** Membrane electrode assemblies were fabricated using a conventional catalyst-coated membrane technique. Nafion 115 membranes purchased from FuelCellsEtc were cut into 3.5 cm × 3.5 cm$^2$ pieces. These membranes were pretreated by soaking in 3% $H_2O_2$ at 80 °C for 1 h, then soaking in 0.5 M $H_2SO_4$ at 80 °C for 1 h and finally soaking in Millipore (18.2 MΩ cm) water at 80 °C for 1 h. The membranes were removed from the water and blotted dry before the catalyst was deposited. Next, a Pt black catalyst (ETEK) and Nafion 117 ionomer solution (Aldrich) were mixed in a 3:1 weight ratio. Separately, an Ir black catalyst (Premetek) and Nafion 117 ionomer solution (Aldrich) were also mixed in a 3:1 weight ratio. The Pt and Ir catalyst/ionomer mixtures were both dispersed in 4:1 volume ratio mixtures of isopropanol and water. The catalyst/ionomer solutions were sonicated for several minutes and then deposited onto opposite sides of the Nafion membranes by spray casting. The Pt catalyst was loaded on the cathode side at 0.5 mg cm$^{-2}$ and the Ir catalyst was loaded on the anode side at 2 mg cm$^{-2}$ over a 2.5 cm × 2.5 cm area for a total device active area of 6.25 cm$^2$. This catalyst-coated membrane was pressed between carbon paper (Sigracet GDL 35BC, Ion Power) on the cathode side and Ti mesh (Dexmet) on the anode side. Two identical assemblies prepared in this manner were loaded into cell assemblies (5 cm$^2$, Fuel Cell Technologies, Inc.), which were maintained at a temperature of 80 °C for all measurements. The two electrolysers were connected in series. Millipore water (18.2 MΩ cm) preheated to 80 °C was fed into the anode side of the first electrolyser; there was no input to the cathode side. The cathode and anode outputs of the first electrolyser were connected to the cathode and anode inputs of the second electrolyser and both outputs of the second electrolyser were collected; thus, the $H_2$ and $O_2$ products could be quantified using a volume displacement Faradaic efficiency measurement apparatus.

**PV-electrolysis system operation.** To construct the PV-electrolysis system, the triple junction PV cell, the two electrolysers and a potentiostat (BioLogic, VMP3) were connected in series as follows: the working electrode port of the potentiostat was connected to the bottom contact of the solar cell, the top contact of the solar cell was connected to the anode of the first electrolyser, the cathode of the first electrolyser was connected to the anode of the second electrolyser and the cathode of the second electrolyser was connected to the counter electrode port of the potentiostat. The potentiostat reference lead was connected to the counter electrode lead so that it could measure the current passing through the closed system; no additional potential was applied. All electrical connections were made with standard copper cables, which introduced negligible resistance compared with other components of the system.

Before the start of the operation, preheated Millipore water was purged with $H_2$ and $O_2$, and pumped into the two electrolysers with a Chem-tech Series 100 pump at a flow rate of 42 ml min$^{-1}$. The solar cell was kept at 25 °C on a water cooler stage and positioned under the multi-sun solar simulator. The distance between the cell and the solar simulator was adjusted so that ∼42 suns of solar concentration was achieved. At this concentration, the solar cell output a short circuit photocurrent of 184 mA (∼583 mA cm$^{-2}$) and aligned the solar cell $I$–$V$ curve for an optimal operation point to match the electrode size and electrolyser capacity.

To begin operation, the shutter of the solar simulator was opened. The system current was recorded continuously by the potentiostat and these data were used to calculate the STH efficiency as a function of time as shown in Fig. 4. The system was run continuously for 48 h without interruption or modification. Periodically throughout the experiment, the gas products from the cathodes and anodes of the electrolysers were collected using volume displacement devices to calculate the Faradaic efficiency. At the end of the 48 h operation, the shutter of the solar simulator was closed.

**Data availability.** The data that support the findings of this study are available from the authors on request.

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

## Acknowledgements

This study presents results from an NSF project (award number CBET-1433442) competitively selected under the solicitation 'NSF 14-15: NSF/Department of Energy Partnership on Advanced Frontiers in Renewable Hydrogen Fuel Production via Solar Water Splitting Technologies', which was co-sponsored by the National Science Foundation, Division of Chemical, Bioengineering, Environmental and Transport Systems (CBET), and the US Department of Energy, Office of Energy Efficiency and Renewable Energy, Fuel Cell Technologies Office. J.D.B. and J.J. acknowledge support from Stanford Graduate Fellowships. L.C.S. acknowledges support from the DARE Doctoral Fellowship provided by the Vice Provost for Graduate Education at Stanford University. J.D.B. and L.C.S. acknowledge support from the National Science Foundation Graduate Research Fellowship Program. We thank Solar Junction Corporation for providing the solar cell samples. We also thank Thomas Hellstern, Pongkarn Chakthranont, Vijay Parameshwaran and Ariel Jackson for helpful discussions.

## Author contributions

J.J., L.C.S. and J.D.B. contributed equally to this work. J.J., L.C.S., J.D.B., Y.H., J.S.H. and T.F.J. conceived the study and designed the experiments. J.J., L.C.S. and J.D.B. performed the experiments. T.B. and Y.C. contributed to PV cell preparation and J.W.D.N. contributed to electrolyser preparation. All authors contributed to analysing the data and writing the manuscript.

## Additional information

**Competing financial interests:** The authors declare no competing financial interests.

