## [Peer Review File · Nature Communications]

Reviewers' comments:

Reviewer #1 (Remarks to the Author):

This paper follows a number of recent reports demonstrating how various independent PV and water electrolyzer modules can be optimally combined to produce high solar to hydrogen conversion efficiency. Under more normal scientific conditions I would suggest that the relatively small improvement here from current record of 24.4% to the 28% demonstrated in this paper (the initial result hardly counts and the authors should remove this from the title!) might not be sufficient to warrant publication in Nature Communications. However, I think that (i) given the huge significance of solar energy storage today and (ii) that there are huge sums of public money being poured into large research projects that are nowhere near this level of efficiency, research such as this indeed has high significance. On that basis I recommend publication. The work of course has been expertly carried out as expected from this group.

Reviewer #2 (Remarks to the Author):

The manuscript reports a solar water splitting system based on a triple junction (3J) solar cell and two series-connected PEM electrolyzers. The work is very well structured and described and, although PV electrolysis is not really novel in itself, the combination of two PEM electrolyzers and a the high effective approach to the water splitting problem is well sought for in the field and would be appropriate in Nature communication as it very well can have a large influence on how researcher approach this problem. Since it is not the price of the device in itself that dominates the price of the hydrogen in the end but instead the total amount of hydrogen gas produced by the device during the life time of the device as well the system/installation cost that is lower for a small area solution, an expensive, small and high efficient device could very well give the cheapest hydrogen in the end. In summary, the series-connected PEM approach, the high conversion in the constructed PEMS as well as the limiting values that can be reached in a real proof-of-concept system I judge to be highly valuable and important to the field and I recommend it to be published.

There are some critical points, however, that I strongly believe should be addressed before publication, which possibly also imply a modification of the title.

1. The authors should include the faradic efficiency as a factor in eq 1.

2. The efficiency is P_{out}/P_{in} as the authors also write for STH in eq S1 in supplementary information. Since the electrolysis is not performed unassisted in this study, the extra energy added to the system to make it run has to be added or at least discussed. Here $P_{in} = \text{Solar energy} + \text{Energy to cool the device to } 25 \text{ }^{\circ}\text{C}$ (Line 91) + Energy to heat the PEM to 80°C (line 102) + Energy of the pump (line 96) enabling the use of a two series-connected PEMS.

Any other temperature lower of the PEM, say 50°C would give lower conversion efficiency as well as if the 3J solar cell was warmer (here, a commonly reported drop of 6% has been reported if not cooled down where this would decrease the STH to 24-25% or lower). The energy of the pump could probably be neglected in a large-scale application but the other energies will at least to an extent scale with the size of the system. Here, a correction of the STH energy should be performed and unfortunately lower the STH conversion efficiency. Another option is that the authors address this and say that this is neglected and instead report the values as the limiting values for assisted PV-electrolysis where the added bias is not considered.

3. The assumption that a photocurrent 42 times the current at 1 sun implies 42 suns is the

intensity at the solar cells has to be verified. If the interconnects have a non-linear behavior above e.g. 30 or 40 suns. It could for example be 44 suns reaching the solar cell but current flow limitations or other non-linear effects in the 3J could give only 42 times the photocurrent. This would then give higher STH efficiency than the actual one. This can be resolved by measuring the actual intensity by e.g. a pyranometer.

4. The (real) intensity at the solar cell surface of course appropriate to use but in a real situation using a solar concentrator, there is an additional issue. Not all light that is concentrated on the outer part of the solar concentrator is coming through and the amount is dependent on the type of concentrator one uses. This would mean that in a real situation the real efficiency would be about 10% lower (lens based concentrator) from losses of incoming light in the solar-concentrator. This is commonly not corrected for even in the solar cell reports of efficiencies so if not corrected for I think it is fair to at least add a comment about this in the manuscript.

Reviewer #3 (Remarks to the Author):

The manuscript presents a nice engineering study of how to match an electrolyzer(s) with a high efficiency solid state solar cell to achieve a high efficiency of solar driven hydrogen evolution. On the one hand it is an obvious exercise with no original scientific content on the other hand it achieves a high conversion efficiency and highlights the competition from the two device approach of PV connected to electrolyzer can be very efficient albeit at rather high cost. Some discussion of the cost of this approach in the text may be appropriate. Considering that two well developed technologies were mated it was a bit disappointing that the output decreased 10% in just 48 hours. I have two specific comments to address if this paper is accepted:

1. I would actually like a revised figure one which now admits that liquid and gas flows are not included to include this in the diagram as a good chemical engineer should do. Given that this diagram may be copied and the text describes this in detail it would be good to show this on the block diagram.
2. There is work from Licht et al in the late 80's and early 90's on this very subject that is not cited. Although I am not a big fan of these papers they are similar and should be cited. Most of the citations can be tracked through the following paper.

J. Phys. Chem. B, Vol. 104, No. 38, 2000

Reviewer #1:

“This paper follows a number of recent reports demonstrating how various independent PV and water electrolyzer modules can be optimally combined to produce high solar to hydrogen conversion efficiency. Under more normal scientific conditions I would suggest that the relatively small improvement here from current record of 24.4% to the 28% demonstrated in this paper (the initial result hardly counts and the authors should remove this from the title!) might not be sufficient to warrant publication in Nature Communications. However, I think that (i) given the huge significance of solar energy storage today and (ii) that there are huge sums of public money being poured into large research projects that are nowhere near this level of efficiency, research such as this indeed has high significance. On that basis I recommend publication.

The work of course has been expertly carried out as expected from this group.”

Response: We thank Reviewer #1 for his or her time in reading the manuscript and providing comments concerning its relevance to the field of solar energy storage. Regarding the reviewer’s comment about the device efficiency, we believe that our STH figure (Figure 4) might not have been sufficiently clear to present our data which then led to a misinterpretation of our results. We certainly agree that any initial transients associated with capacitive current should not count towards a measured or reported STH efficiency. However, we measured greater than 30% STH for the first 20 hours of device testing which we think is more than an “instantaneous” or “initial” result and deserves to be highlighted, including in the title. The average STH efficiency for the entire 48.3 hours was 29.9% and even the average STH for the last 20 minutes of testing was 28.5%, which is greater than the 28% cited by the reviewer. Respectfully, we believe that disregarding all of the data except for the last 20 minutes of measuring time is misleading. Additionally, many previous literature reports only show measurements for 24 hours or less; by this metric our average STH efficiency for the first 24 hours is 30.4%. To improve the clarity of reporting our device performance, we have included an inset in Figure 4 (which is also shown below) which highlights a narrower y-axis to show that the STH efficiency is at or above 30% for 24+ hours. We have also edited the text in the manuscript to include additional STH efficiency metrics which we hope will be helpful for the reader. Regarding the title of the manuscript, for the reasons above we believe that maintaining a mention to the “efficiency over 30%” is still appropriate, though we are open to alternative suggestions from the Editor.

Reviewer #2:

“The manuscript reports a solar water splitting system based on a triple junction (3J) solar cell and two series-connected PEM electrolyzers. The work is very well structured and described and, although PV electrolysis is not really novel in itself, the combination of two PEM electrolyzers and a the high effective approach to the water splitting problem is well sought for in the field and would be appropriate in Nature communication as it very well can have a large influence on how researcher approach this problem. Since it is not the price of the device in itself that dominates the price of the hydrogen in the end but instead the total amount of hydrogen gas produced by the device during the life time of the device as well the system/installation cost that is lower for a small area solution, an expensive, small and high efficient device could very well give the cheapest hydrogen in the end. In summary, the series-connected PEM approach, the high conversion in the constructed PEMS as well as the limiting values that can be reached in a real proof-of-concept system I judge to be highly valuable and important to the field and I recommend it to be published.

There are some critical points, however, that I strongly believe should be addressed before publication, which possibly also imply a modification of the title.”

Response: We thank Reviewer #2 for his or her time in reading the manuscript and providing a critical analysis of our work that has allowed us to improve the manuscript. We have responded in detail to the reviewer’s points below, noting modifications to the manuscript where applicable.

1. The authors should include the faradic efficiency as a factor in eq 1.

Response: We have modified equation 1 to include the Faradaic efficiency as a factor. As discussed in the supporting information, on the basis of our product quantification measurements and calculations, we concluded that the Faradaic efficiencies for H₂ and O₂ production were effectively 100%. While we measured Faradaic efficiencies of less than 100%, we believe this is due to measurement error and not due to a real

diversion of charge passed into side reactions, as is explained in great detail in the supporting information. As a result, this modification does not affect our reported STH values.

2. The efficiency is P_{out}/P_{in} as the authors also write for STH in eq S1 in supplementary information. Since the electrolysis is not performed unassisted in this study, the extra energy added to the system to make it run has to be added or at least discussed. Here $P_{in} = \text{Solar energy} + \text{Energy to cool the device to } 25^{\circ}\text{C (Line 91)} + \text{Energy to heat the PEM to } 80^{\circ}\text{C (line 102)} + \text{Energy of the pump (line 96)}$ enabling the use of a two series-connected PEMs.

Any other temperature lower of the PEM, say 50°C would give lower conversion efficiency as well as if the 3J solar cell was warmer (here, a commonly reported drop of 6% has been reported if not cooled down where this would decrease the STH to 24-25% or lower). The energy of the pump could probably be neglected in a large-scale application but the other energies will at least to an extent scale with the size of the system. Here, a correction of the STH energy should be performed and unfortunately lower the STH conversion efficiency. Another option is that the authors address this and say that this is neglected and instead report the values as the limiting values for assisted PV-electrolysis where the added bias is not considered.

Response: In general, we agree with the reviewer that any additional energy inputs into the device are important to consider, and that for a full commercial system such measurements should be factored into the STH efficiency calculation. However, for this proof-of-concept laboratory scale device, such energy inputs are not only difficult to measure accurately, they also might not be relevant for a scaled-up system operating at the commercial level. For instance, the tightly packaged nature of commercial-scale electrolyzers leads to nearly adiabatic operation, so the overpotential losses at the anode and cathode will naturally heat the device significantly without requiring an additional source of heat. For these and related reasons considering the differences between commercial devices and the proof-of-concept laboratory scale device we report here, we chose to neglect these energy inputs in the STH calculation. Instead, as the reviewer suggested, we added several paragraphs of discussion to the manuscript to provide a broader and deeper discussion of these important points. We listed these specific energy inputs, explained why we have neglected them, and clarified that our STH efficiency represents a limiting upper bound value assuming that these additional energy inputs are negligible.

3. The assumption that a photocurrent 42 times the current at 1 sun implies 42 suns is the intensity at the solar cells has to be verified. If the interconnects have a non-linear behavior above e.g. 30 or 40 suns. It could for example be 44 suns reaching the solar cell but current flow limitations or other non-linear effects in the 3J could give only 42 times the photocurrent. This would then give higher STH efficiency than the actual one. This can be resolved measuring the actual intensity by e.g. a pyranometer.

Response: We thank the reviewer for pointing out the possibility of error in our efficiency calculation. Nevertheless, we believe our method of determining the simulated solar concentration is accurate. This method is routinely used and is considered a standard approach in the solar cell community, particularly for III-V cells.

The photovoltaic cell we used in this study is a commercial PV, and has been thoroughly characterized by the manufacturer. Like other multi-junction III-V PV cells, the manufacturer found the photocurrent response of this cell to be directly proportional to the illumination intensity. Thus, the short circuit current density ratio is a very accurate method for measuring the illumination intensity; the spectrum used in standard reporting conditions (SRC) for concentrator solar cells is defined as the AM1.5D spectrum scaled by the ratio of the short-circuit current (J_{sc}) at high irradiance to the 1-sun value. The cell we employed in our study was actually designed to operate under extremely high solar concentrations, near 1000 suns, and as such the metal contacts are optimized to accommodate high currents. Under 42 suns of concentration the resistive effect is negligible. We also did not use any lenses or other optics in front of the cell, so there are no nonlinear effects due to optics,

such as chromatic aberration. Therefore, it is a common practice to determine the simulated solar concentration from the J_{SC} ratio, as described in the following references, which have now been cited in the manuscript:

Osterwald, C., Wanlass, M., Moriarty, T., Steiner, M. & Emergy, K. in *IEEE 40th Photovoltaic Specialist Conference (PVSC)* 2616-2619 (Denver, CO, 2014).

Osterwald, C. *Translation of Device Performance Measurements to Reference Conditions*. Vol. 18 269-279 (1986).

We also modified the corresponding paragraph in the Methods section to clarify the method we used to calculate the simulated solar concentration. We thank the reviewer again for this thoughtful suggestion.

4. The (real) intensity at the solar cell surface of course appropriate to use but in a real situation using a solar concentrator, there is an additional issue. Not all light that is concentrated on the outer part of the solar concentrator is coming through and the amount is dependent on the type of concentrator one uses. This would mean that in a real situation the real efficiency would be about 10% lower (lens based concentrator) from losses of incoming light in the solar-concentrator. This is commonly not corrected for even in the solar cell reports of efficiencies so if not corrected for I think it is fair to at least add a comment about this in the manuscript.

Response: The reviewer raises another excellent point. We agree that optical losses due to imperfect reflection or transmission from the optical components of the concentrator can result in efficiency losses in concentrated PV systems that are illuminated with real sunlight. In our laboratory scale demonstration system, we did not use any lenses or mirrors to concentrate the simulated solar light, so there are no optical losses to measure. However, we added discussion to the manuscript to clarify that these losses are not considered in our analysis, and that they have the potential to reduce the STH efficiency values of PV-electrolysis systems illuminated with actual, concentrated solar light.

Reviewer #3:

“The manuscript presents a nice engineering study of how to match an electrolyzer(s) with a high efficiency solid state solar cell to achieve a high efficiency of solar driven hydrogen evolution. On the one hand it is an obvious exercise with no original scientific content on the other hand it achieves a high conversion efficiency and high lights the competition from the two device approach of PV connected to electrolyzer can be very efficient albeit at rather high cost. Some discussion of the cost of this approach in the text may be appropriate. Considering that two well developed technologies were mated it was a bit disappointing that the output decreased 10% in just 48 hours. I have two specific comments to address if this paper is accepted:”

Response: We thank reviewer #3 for his or her time in reading the manuscript and providing a thoughtful assessment. We agree with the reviewer that this work is a useful contribution to the solar and renewable energy communities as this device was engineered to push the bounds of efficiency for electrochemical H_2 production as a demonstration of what is currently, technologically possible. Below we address the reviewer’s specific comments in detail and make corresponding changes to the manuscript.

In regards to the reviewer’s comment about the cost of the device and the observed degradation, we have edited the manuscript to address these concerns. In this work, we have focused on engineering a high efficiency device because analyses of commercial scale electrochemical hydrogen production plants suggest that STH efficiency has a very large effect on the final cost of produced H_2 , potentially an even larger effect than cost of the absorber and catalyst materials or the device lifetime. However, we agree with the reviewer that subsequent work pursuing development of high performance, low cost electrolyzer systems would be greatly beneficial for

the eventual commercialization of this technology. Such efforts are largely underway by PEM electrolyzer manufacturers such as Giner and Proton Onsite.

In order to achieve the very high efficiency demonstrated in this study, we optimized the current match between the solar cell and the electrolyzers, such that the system operated as close as possible to the maximum power point of the solar cell IV curve. Therefore, even a small amount of degradation of the electrolyzer is expected to cause a measurable decrease in overall device performance. As mentioned in the manuscript, drift in the output of the solar simulator lamp's output also accounted for a significant fraction of the observed decrease in STH efficiency. We note that our laboratory scale demonstration device is constructed from components that are available for use on a larger scale. Commercial photovoltaic cells as well as commercial PEM electrolyzers have both achieved very good long term stability. Therefore, we believe our results still highlight the great potential of this high efficiency approach to solar H₂ production even though our proof-of-concept device did experience some decrease in STH over the 48-hour experiment. We have edited the manuscript text to include additional discussion comparing our lab scale device to potential commercial devices that could be designed using similar components.

1. I would actually like a revised figure one which now admits that liquid and gas flows are not included to include this in the diagram as a good chemical engineer should do. Given that this diagram may be copied and the text describes this in detail it would be good to show this on the block diagram.

Response: We thank the reviewer for this useful suggestion and we have updated the block flow diagram to include the liquid and gas flow streams. We agree that the more complete figure will be helpful for readers.

2. There is work from Licht et al in the late 80's and early 90's on this very subject that is not cited. Although I am not a big fan of these papers they are similar and should be cited. Most of the citations can be tracked through the following paper: J. Phys. Chem. B, Vol. 104, No. 38, 2000

Response: We thank the reviewer for pointing out this highly relevant collection of work. We have included citations of this work in our manuscript to provide greater context for our work.

We thank the reviewers for taking the time to carefully consider our manuscript and provide constructive feedback.

Sincerely,

Thomas F. Jaramillo
Associate Professor
Dept. of Chemical Engineering, Stanford University
Email: jaramillo@stanford.edu, Web: jaramillogroup.stanford.edu
Phone: +1-650-498-6879